# The Past and Present of Discharge Capacity Modeling for Spillways—A Swedish Perspective

**James Yang** [1,2,*], **Patrik Andreasson** [1,3], **Penghua Teng** [2] **and Qiancheng Xie** [3]

[1] Vattenfall AB, Research & Development (R & D), Hydraulic Laboratory, 81426 Älvkarleby, Sweden; patrik.andreasson@vattenfall.com

[2] Division of Resources, Energy & Infrastructure, Royal Institute of Technology, 10044 Stockholm, Sweden; teng.penghua@byv.kth.se

[3] Division of Fluid & Experimental Mechanics, Luleå University of Technology, 97187 Luleå, Sweden; qiancheng.xie@ltu.se

[*] Correspondence: james.yang@vattenfall.com; Tel.: +46-8-739-5000

**Abstract:** Most of the hydropower dams in Sweden were built before 1980. The present dam-safety guidelines have resulted in higher design floods than their spillway discharge capacity and the need for structural upgrades. This has led to renewed laboratory model tests. For some dams, even computational fluid dynamics (CFD) simulations are performed. This provides the possibility to compare the spillway discharge data between the model tests performed a few decades apart. The paper presents the hydropower development, the needs for the ongoing dam rehabilitations and the history of physical hydraulic modeling in Sweden. More than 20 spillways, both surface and bottom types, are analyzed to evaluate their discharge modeling accuracy. The past and present model tests are compared with each other and with the CFD results if available. Discrepancies do exist in the discharges between the model tests made a few decades apart. The differences fall within the range −8.3%–+11.2%. The reasons for the discrepancies are sought from several aspects. The primary source of the errors is seemingly the model construction quality and flow measurement method. The machine milling technique and 3D printing reduce the source of construction errors and improve the model quality. Results of the CFD simulations differ, at the maximum, by 3.8% from the physical tests. They are conducted without knowledge of the physical model results in advance. Following the best practice guidelines, CFD should generate results of decent accuracy for discharge prediction.

**Keywords:** spillway; bottom outlet; design flood; discharge capacity; model tests; computational fluid dynamics (CFD)

## 1. Hydropower Development in Sweden

In Sweden, an abundance of river streams that connect with more than 100,000 lakes gives the country's landscape a unique character and beauty. The first hydro-electrical power station was put into operation in 1882. The pioneering construction of large dam facilities dated back to as early as the 1910s, during which Porjus, Trollhättan and Älvkarleby, as the first large dams, were built.

The large-scale development of hydropower started immediately after the 2nd World War and reached its peak during the 1950s and 1960s. Figure 1 shows the annually installed capacity over the years of the hydropower development. The economically feasible potential of hydropower in the country amounts to approximately 95 terawatt hours (TWh), about two thirds of which have already been tapped. The remaining part is left now in its natural conditions. Four major rivers, i.e., Vindelälven, Pite älv, Kalix älv and Torne-Muonio älv, are protected by the law from development.

The country's total installed generating capacity totals up to 16 200 MW. Depending upon the precipitation and reservoir water storage, the power production varies between 60 and 70 TWh in a normal year, accounting for 40–50% of the total power production in the country [1].

There are approximately 1000 hydropower dams of varying size and age (one power station has two or more dams). Most of the rivers with high dams are in Northern Sweden. According to the International Commission on Large Dams (ICOLD) definition of large dams (dam height ≥ 15 m), there are approximately 190 large dams in the country, in which about 80% are embankment dams with impervious core.

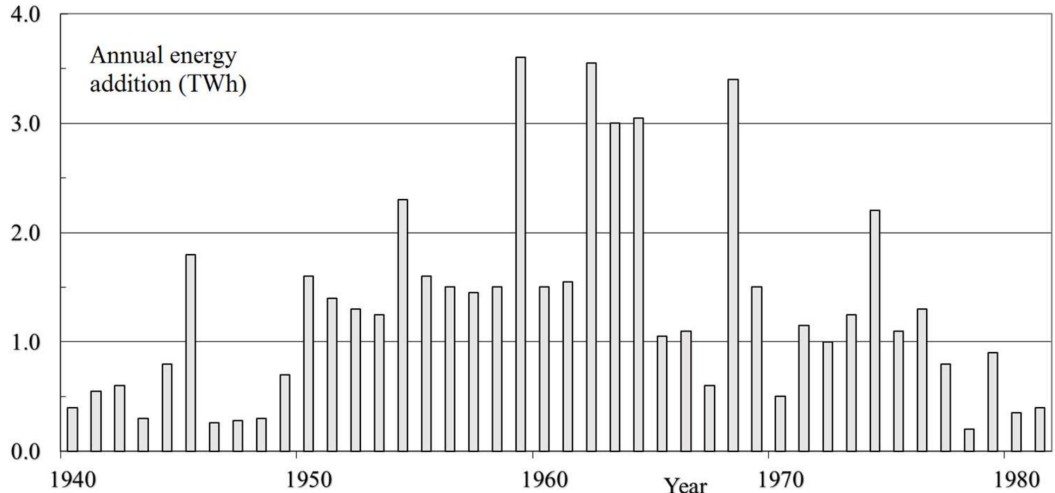

**Figure 1.** Annual addition of power capacity in Sweden.

The largest river is the Lule älv river with 15 large dams, ten of which are situated north of the Arctic Circle. They account for approximately 10% of the hydroelectric production in the country. The highest dam is Trängslet on the Dalälven river, a 122 m embankment dam completed in 1960. The largest power plant is the 945 MW Harsprånget on the Lule älv river, operating at a 107 m head and a 1040 $m^3$/s turbine flow rate. Vänern on the Göta älv river is the largest hydropower reservoir, with a 9400 $Mm^3$ active storage volume.

During the dam constructions, dam safety was not a subject of state regulatory surveillance; there were no national directives that governed the dam design, construction, and supervision. The responsibility rested totally with the dam developer. The design flood of a dam was often finalized via multiplication of the highest historical flood with a safety factor. For some small or even medium dams, the frequency analysis method was used, in which the 1000-year flood, obtained through extrapolation, was treated as the design flood. The spillways were sized according the dam-safety guidelines that were available at the time of construction. The hydrological methods previously used to determine the design flood were inaccurate [2]. During the past decades, operations of the dams have also evidenced the insufficiency of the previous practice for the spillway capacity determination.

In a Swedish perspective, the paper reviews the history of physical hydraulic modeling for hydropower dams and discusses the needs for dam rehabilitations that lead to renewed model tests. The purpose of the study is threefold. (1) To compare the accuracy of the discharge results between the model tests performed many decades apart. Do we build better physical models and make better tests nowadays? (2) To compare CFD results with the physical test ones. What is the accuracy? Can CFD predict reasonably accurate as a design tool without physical model testing? (3) To explain the sources of errors associated with both the physical and numerical modeling.

## 2. History of Physical Modeling of Dams in Sweden

For a dam, its physical hydraulic modeling aims at verification of the design layout, solution of already known problems and improvement in the design. Perhaps the most significant benefit of the model tests is the identification of unexpected problems that are disregarded in the design process. Issues such as unfavorable flow patterns, occurrence of vortices and sediment movement leading to deposition are readily observed in a physical model. Most of the dams were built between 1945 and 1980. The hydraulic design of many of the spillways was solely evaluated through hydraulic modeling prior to or during the construction. In addition to examining the overall layout, determination of the spillway discharge capacity was always a major task.

Historically, the physical model tests were performed at three hydraulic laboratories, i.e., the Royal Institute of Technology (KTH), Stockholm; Chalmers University of Technology (CTH), Gothenburg, and the power producer Vattenfall (previously the Swedish State Power Board), Älvkarleby. Along with the decreasing need for hydraulic modeling activities after the peak period of the hydropower development, the CTH and KTH laboratories were successively closed. Vattenfall's hydraulic laboratory is the only testing facility that remains operational in the country.

The revised criteria for flood determination in the country are the basis for the revision of the dam-safety guidelines. The criteria give rise to a considerable increase in the magnitude of design floods. As a result, most of the existing dams, especially high-hazard ones, need to be rebuilt or reinforced to meet the higher safety requirements. Obviously, this becomes an issue that involves large capital investment and takes many years to accomplish.

In connection with the upgrading requirements, many dam spillways have been re-tested in Vattenfall´s laboratory. The proposed rebuilding measures of the dams are examined and evaluated, so that risks for damages in the dam body and its foundation that could endanger the dam safety are eliminated for flood discharges up to the design-flood magnitude. Bergeforsen and Ajaure are among the first dams that were re-evaluated during 1997–2000. As of 2018, some 40 existing dams have been evaluated in the laboratory [3–5].

Höljes is an 80 m high embankment dam on the Klarälven river in South Sweden. The facility was originally equipped with two gated overflow openings. Figure 2 shows its scale models built at CTH during 1958 and at Vattenfall during 2008 [6–8]. The model scales are 1:60 and 1:50, respectively. In the renewed model tests, a few rebuilding options, including a tunnel spillway and an ungated overflow weir stretching into the reservoir, are evaluated to raise the spillway capacity. The final solution is a new 17 m wide spillway opening with an upward-going radial gate. The abandoned timber flume located to the right of the existing openings is removed to give place to the new gate. To accommodate the increased discharge, the width of the spillway channel downstream is doubled, and the stilling basin is also enlarged. The dam spillway before and after the 2016 rehabilitation is shown in Figure 3.

Statistics has been conducted on the scale models built during the past two decades at Vattenfall's hydraulic laboratory, showing that the models are 6–25 m long, 5–15 m wide and 0.5–2.0 m high. Depending upon the complexity of the hydraulic issues raised in the dam refurbishments, the model construction and tests that follow involve costs in a range of 50 000–300 000 US$. In accordance with the dam-safety guidelines and the performed model tests, Ajaure, a 45 m embankment dam with an active storage of 200 Mm$^3$, was the first dam refurbished some 20 years ago. The upgrade measures are a combination of the dam crest raise including its impervious core of moraine and modification of one bottom outlet into a gated overflow spillway [3,9]. The augmented spillway capacity reduces the extent of the dam crest heightening and accordingly the total engineering costs.

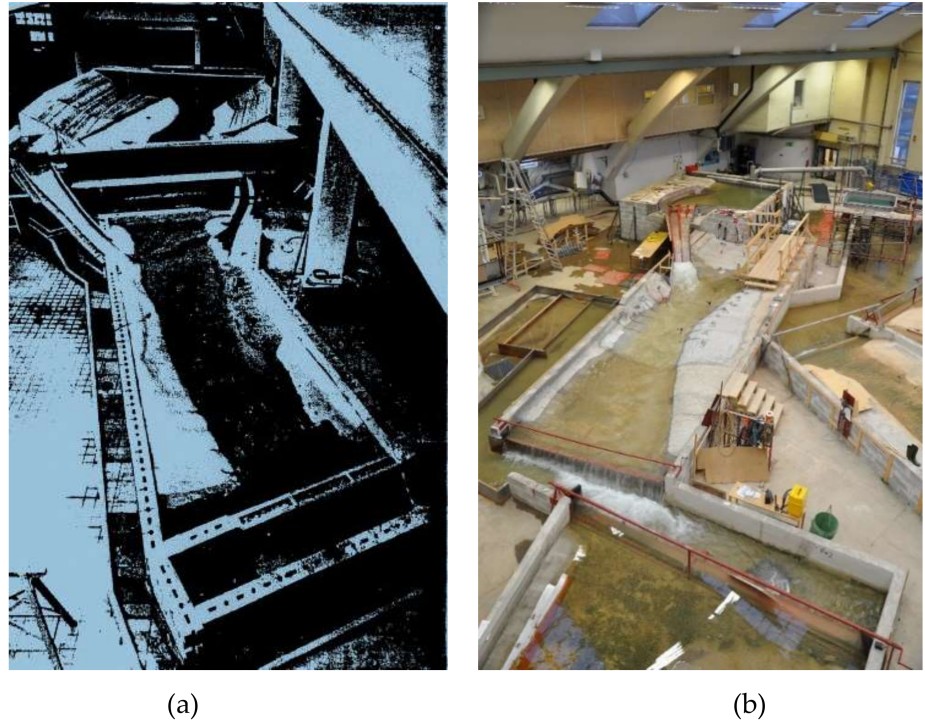

Figure 2. Höljes dam spillway, two physical scale models constructed five decades apart: (**a**) 1958 at Chalmers University of Technology (CTH); (**b**) 2008 at Vattenfall's hydraulic laboratory.

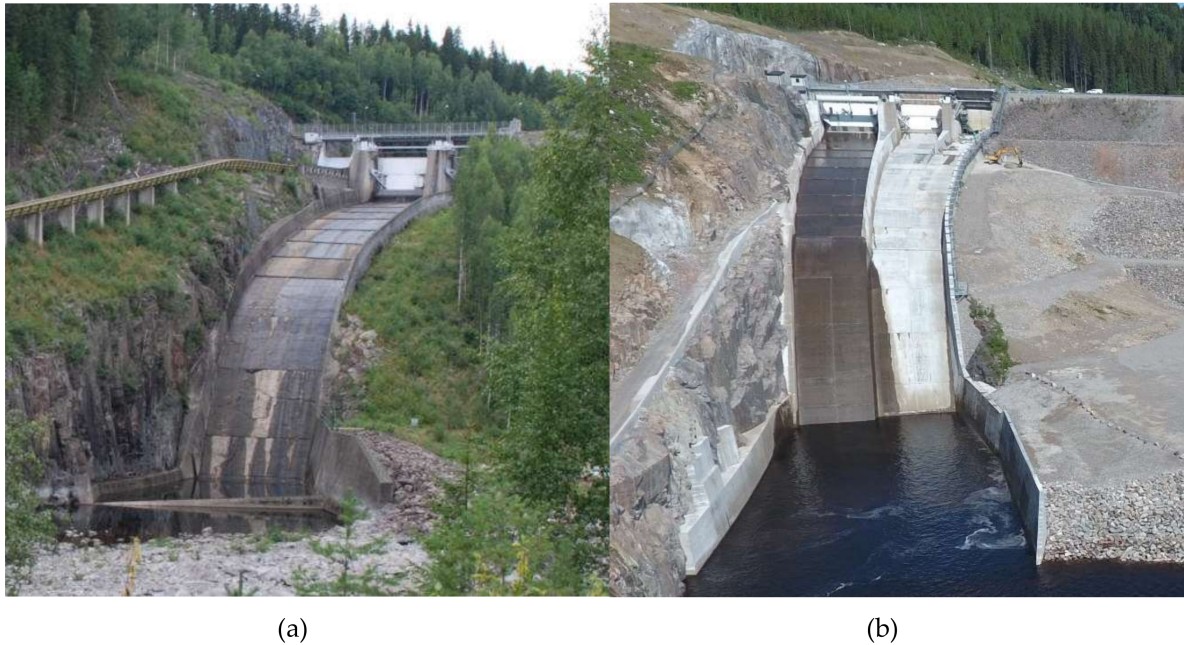

Figure 3. Höljes dam spillway: (**a**) Original layout with two gated openings and a timber flume (© Fortum Generation AB); (**b**) After refurbishment, with a 17 m new gated opening added adjacent to the existing ones (© Hydro Terra AB).

## 3. Needs for Dam Refurbishment

Based on the new flood determination criteria, many existing dams are found to have higher design floods than constructed for. The recent revision of the flood criteria was made in 2015 [10]. The design flood for a dam refers normally to the required spillway discharge capacity at the design reservoir water level (DRWL). For 15 selected dams on seven rivers, Table 1 shows a list of the original

design floods (denoted as $Q_0$, m$^3$/s) that the dams were originally designed for and the increment in percentage (denoted as $\xi$) of the revised floods, where $Z$ (m) refers to the maximum structural height of a dam. Abbreviations R, E, G and Bu denote the rock-fill, earth-fill, gravity, and buttress dam types, respectively. The dams are built between 1944 and 1983; embankment dams of rock- and earth-fill are a common type in Sweden.

Counting in many other dams, the statistics shows that the revised design floods are typically 20–50% higher [4,11]. This implies that most of the existing spillways in the country are undersized. Only a limited number of spillways in the country have higher discharge capacity than their design floods.

The dam-safety guidelines [12] stipulate that a dam should discharge and withstand the revised design flood without jeopardizing its structural integrity. The rebuilding measures include modification of the existing spillway, construction of a new spillway, raising the dam crest and its impervious core, repair, or replacement of erosion protection upstream of embankment dams, structural reinforcement of the dam body, re-shaping and strengthening of waterways and enlargement of energy dissipator including stilling basin and plunge pool. The spillway modification refers to width enlargement, lowering of sill elevation and change of a bottom outlet to a gated overflow opening. It is common that several rebuilding measures are combined to achieve a technically feasible and cost-effective solution [4,9].

One of the primary goals of the renewed physical modeling is the determination of discharge capacity for the spillway in question. CFD modeling is performed for the same purpose, either separately or in parallel with the model tests. The information is then used as input in the design of the rebuilding measures. The new models also provide discharge data for the original spillway layouts. This gives the possibility to compare the test results with those from the dam construction periods. It is found that the differences, significant in some cases, in the discharge capacity exist between the present and past model test results and between the numerical and physical modeling results, which obviously has an impact on the rehabilitation options and costs. Questions and discussions arise then about the accuracy of both methods, the reason for the discrepancy and governing factors in modeling.

**Table 1.** Higher design floods as required by the updated flood criteria in Sweden.

| No. | River Name | Dam Name | Dam Type | Year of Completion | $Z$ (m) | $Q_0$ (m$^3$/s) | $\xi$ (%) |
|---|---|---|---|---|---|---|---|
| 1 | | Ajaure | R | 1967 | 45 | 950 | 40 |
| 2 | Ume älv | Stornorrfors | Bu/R | 1958 | 20 | 3300 | 35 |
| 3 | | Rusfors | E | 1962 | 22 | 1625 | 27 |
| 4 | Indalsälven | Midskog | G/E | 1944 | 27 | 2300 | 35 |
| 5 | | Bergeforsen | E | 1955 | 29 | 2300 | 45 |
| 6 | Ångermanälven | Stenkullafors | E | 1983 | 30 | 1250 | 40 |
| 7 | | Edensforsen | E | 1956 | 19 | 1400 | >40 |
| 8 | Skellefte älv | Gallejaur | R/E | 1964 | 55 | 700 | 20 |
| 9 | Klarälven | Höljes | R/E | 1961 | 80 | 1600 | >25 |
| 10 | Ljusnan | Långströmmen | R/E | 1961 | 28 | 1670 | 50 |
| 11 | | Halvfari | E/Bu | 1978 | 43 | 650 | 100 |
| 12 | | Letsi | R | 1967 | 85 | 1500 | 25 |
| 13 | Lule älv | Porsi | E | 1962 | 40 | 2700 | 15 |
| 14 | | Vittjärv | G | 1974 | 15 | 2200 | 50 |
| 15 | | Boden | E | 1971 | 21 | 2800 | 20 |

## 4. Examined Spillways

To clarify the issue, more than 20 dams are reviewed to examine their spillway discharge capacity. A total of 19 gated overflow spillways (including the Vatnsfell spillway on Iceland) and 4 bottom outlets are included. The present data refer to the results from the renewed laboratory model tests and CFD studies from the past two decades. The hydraulic conditions downstream of the dams are not an issue of the study.

Tables 2 and 3 list the gated overflow spillways and bottom outlets included in the comparison study. For the former, parameters $B$ (m) and $H$ (m) denote total opening width and water head above the sill elevation. For the latter, $a$ (m), $b$ (m) and $H_1$ (m) refer to outlet height, total net width and water head above the invert elevation at the gate position. For each outlet, a short conduit section exists upstream of the segment gate; the outflow is free at the full gate opening, without any tailwater effects, which is a typical Swedish condition. For both types, the water head ($H$ or $H_1$) is measured from the reservoir's undisturbed water surface at the DRWL.

**Table 2.** Gated overflow spillways included.

| No. | Dam Name | Year of Completion | $Z$ (m) | No. of Openings | $B$ (m) | $H$ (m) |
|---|---|---|---|---|---|---|
| 1 | Bergeforsen existing spillway | 1955 | 35 | 3 | 45 | 9.25 |
| 2 | Bergeforsen new spillway | 2014 | 35 | 1 | 15 | 10.25 |
| 3 | Boden | 1971 | 21 | 3 | 45 | 10.94 |
| 4 | Edensforsen | 1956 | 19 | 4 | 58.22 | 4.00–7.70 |
| 5 | Gallejur | 1965 | 15 | 2 | 24 | 6.40 |
| 6 | Halvfari | 1978 | 43 | 2 | 12 | 10.00 |
| 7 | Harsprånget | 1952 | 50 | 3 | 60 | 7.60 |
| 8 | Höljes | 1961 | 80 | 2 | 28 | 8.70 |
| 9 | Laxede | 1962 | 24 | 3 | 45 | 10.44 |
| 10 | Letsi | 1967 | 85 | 2 | 30 | 9.70 |
| 11 | Ligga | 1954 | 35 | 3 | 60 | 7.61 |
| 12 | Långbjörn | 1960 | 30 | 3 | 45 | 7.00 |
| 13 | Långströmmen new spillway | 2018 | 28 | 1 | 18 | 6.55 |
| 14 | Midskog | 1944 | 27 | 4 | 58 | 6.25 |
| 15 | Porsi | 1962 | 40 | 3 | 45 | 10.40 |
| 16 | Torpshammar | 1943 | 25 | 2 | 12 | 3.50 |
| 17 | Stenkullafors | 1983 | 30 | 2 | 20 | 10.00 |
| 18 | Stornorrfors | 1958 | 25 | 3 | 57 | 9.60 |
| 19 | Vatnsfell (Iceland) | 2000 | 30 | 1 | 50 | 2.10 |

**Table 3.** Bottom outlets included.

| No. | Dam Name | Year of Completion | $Z$ (m) | No. of Openings | $a$ (m) | $b$ (m) | $H_1$ (m) |
|---|---|---|---|---|---|---|---|
| 1 | Ajaure | 1967 | 45 | 2 | 10.40 | 10.00 | 16.00 |
| 2 | Satisjaure | 1966 | 30 | 2 | 6.50 | 8.00 | 23.00 |
| 3 | Storfinnforsen | 1953 | 39 | 1 | 3.65 | 5.50 | 37.00 |
| 4 | Torpshammar | 1943 | 25 | 2 | 2.50 | 12.00 | 18.60 |

## 5. Present versus Past Laboratory Tests

Most of the past laboratory tests were made prior to or during the construction time. The present testing is solely due to the ongoing dam refurbishments as imposed by the dam-safety guidelines [12]. All the models are based on the Froude law of similarity, with the effects of viscosity and surface tension checked in the present study. The most common model scales, denoted as 1:$\lambda$, are $\lambda$ = 40–60.

The relative difference in the discharge capacity of a spillway is defined as

$$\alpha \ (\%) = (Q_{\text{present}} - Q_{\text{past}})/Q_{\text{past}} \tag{1}$$

in which $Q_{\text{present}}$ (m$^3$/s) and $Q_{\text{past}}$ (m$^3$/s) refer, from the present and past model tests, to the flow discharges at the same reservoir water level. Table 4 compares the $Q_{\text{present}}$ and $Q_{\text{past}}$ results for 18 spillways [13–22]. All the values refer to the discharge capacity data that correspond to the DRWL. Due to the fragmental documentation, it is not possible to trace the model scales of the past studies, which is the very reason they are not listed in the table.

As seen from Table 4, for the 18 overflow spillways included, the $\alpha$ values vary between −6.0 and +11.2. A positive $\alpha$ value indicates a higher discharge capacity from the present model study. Obviously, both lower and higher results are obtained from the present tests. The accuracy of the physical hydraulic modeling, if correctly performed, is usually in the range ±(2–4)%. Most of the

laboratory test results, both present and past, fall within this range, while a few cases show somewhat larger discrepancies.

**Table 4.** Spillway discharges from the past and renewed physical model tests.

| Dam Name | Past Model Study | | Present Model Study | | | $\alpha$ (%) |
|---|---|---|---|---|---|---|
| | Year | $Q_{past}$ (m³/s) | Year | $\lambda$ | $Q_{present}$ (m³/s) | |
| Ajaure | 1967 | 1020 | 2000 | 50 | 935 | −8.3 |
| Bergeforsen existing spillway | 1955 | 2300 | 1998 | 50 | 2400 | 4.3 |
| Boden | 1971 | 2760 | 2010 | 50 | 2595 | −6.0 |
| Edensforsen | - | 1350 | 2010 | 50 | 1410 | 4.4 |
| Gallejaur | 1962 | 724 | 2004 | 40 | 720 | −0.6 |
| Halvfari | - | 675 | 2007 | 40 | 650 | −3.7 |
| Harsprånget | 1980 | 2340 | 2003 | 60 | 2600 | 11.1 |
| Höljes | 1958 | 1160 | 2008 | 50 | 1290 | 11.2 |
| Laxede | 1962 | 2615 | 2007 | 60 | 2825 | 8.0 |
| Letsi | 1967 | 1555 | 2002 | 50 | 1525 | −1.9 |
| Ligga | 1980 | 2339 | 2005 | 50 | 2225 | −4.9 |
| Långbjörn | 1986 | 1554 | 2006 | 100 | 1585 | 2.0 |
| Midskog | 1942 | 2305 | 2002 | 50 | 2305 | 0 |
| | 1992 | 2375 | | | | −2.9 |
| Porsi | 1961 | 2680 | 2002 | 50 | 2777 | 3.6 |
| Satisjaure | 1962 | 810 | 2003 | 50 | 790 | −2.5 |
| Stenkullafors | 1975 | 1240 | 2003 | 50 | 1250 | 0.8 |
| Storfinnforsen | 1950 | 453 | 2008 | 30 | 449 | −1.0 |
| Stornorrfors | 1950s | 3200 | 2003 | 100 | 3275 | 2.3 |

For the Harsprånget and Höljes spillways, the present tests give >10% higher discharge. This has a bearing on the evaluation of rebuilding options. To find which result is more reliable, both the spillway layout and reservoir topography are compared. However, no obvious difference is found. The present study is based on the echo-sounding made a few years ago; the terrain in front of the spillway might have changed somewhat due to sedimentation. As sediment is of very limited amount in the Swedish rivers, the slight modification in the bed form cannot account for the large discrepancy.

When compared with the past tests, too low or too high capacity is always questioned by the dam owner and needs to be explained. The reason for the discrepancies can be sought in several aspects that follow.

## 6. Source of Errors in Model Tests

Hydraulic modeling, either physical or numerical, requires the same information as basis. This includes flow data, reservoir topography, geometrical layout of dam and spillway and any other structures that affect the discharge. In terms of modeling accuracy, the difference is that a physical model focuses mainly on construction quality and measurement equipment, while a numerical model lays emphasis on grid quality, turbulence model and numerical scheme.

### 6.1. Reservoir Size and Topography

For a typical model in Vattenfall's laboratory, its length (in the main flow direction) and width of the reservoir area are 14–20 times and 12–15 times the width of the spillway structure. The model size should be large enough, so that the approach flow to the spillway, including the side contractions, is reasonably reproduced, without noticeable boundary effects. This is necessary for the modeling accuracy of an overflow spillway, either gated or ungated. A bottom outlet usually features larger water heads; the upstream size plays relatively a minor role.

The reservoir bed roughness is a geometrical factor that must be reproduced. A few decades ago, a model was formed with the help of several cross-sections, usually at an interval of 15–30 cm.

Its bed surface was made of concrete and finished manually, implying that the surface irregularities were mostly smoothed out. Most of our recent models are made with the milling technique through computer numerical control. By adjusting control parameters such as feeding speed, cutter type and diameter, the required surface roughness is easily realized over a large reservoir area. This leads to a higher modeling accuracy and savings in terms of both construction time and material costs. Figure 4 shows the recent models of the Stornorrfors and Lilla Edet dams produced with the milling technique.

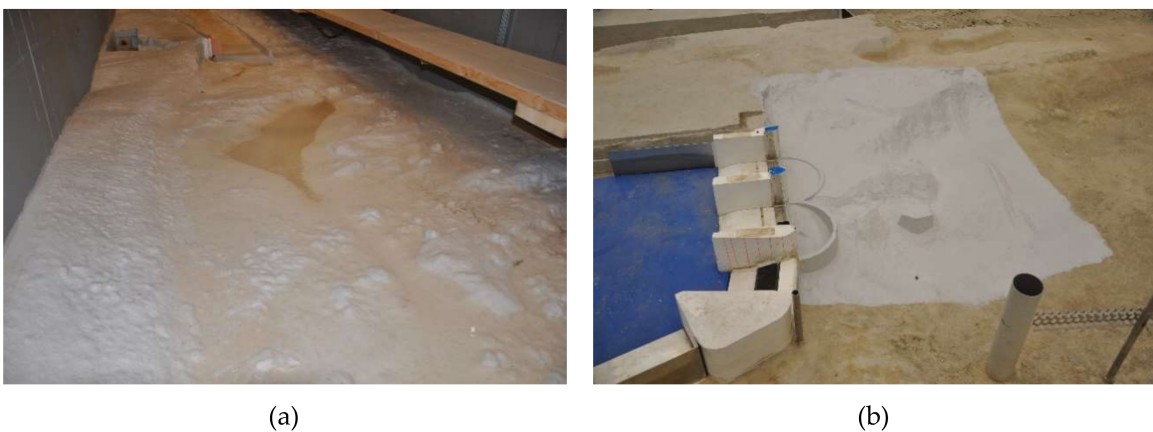

| (a) | (b) |

**Figure 4.** Models produced with the milling technique: (**a**) Stornorrfors, $\lambda = 50$; (**b**) Lilla Edet, $\lambda = 50$.

*6.2. Error Estimations in Model Tests*

For free discharge in an overflow spillway, the discharge $Q$ is expressed as

$$Q = C \times B \times H^{3/2} \tag{2}$$

where $C$ = discharge coefficient dependent on the spillway layout, upstream flow conditions etc. It is usually determined experimentally, and its relative error is written as

$$\frac{\Delta C}{C} = \sqrt{\left(\frac{\Delta Q}{Q}\right)^2 + \left(\frac{\Delta B}{B}\right)^2 + \frac{9}{4}\left(\frac{\Delta H}{H}\right)^2} \tag{3}$$

where $\Delta Q/Q$, $\Delta B/B$ and $\Delta H/H$ refer to the relative error of $Q$, $B$ and $H$. For a submerged bottom outlet with free outflow, $Q$ is written as

$$Q = C_1 \times A \times H_1^{1/2} \tag{4}$$

where $C_1$ = discharge coefficient (dependent on the outlet configuration, upstream flow conditions etc.) and $A = a \times b$, the bottom outlet's total area. The relative error is given by

$$\frac{\Delta C_1}{C_1} = \sqrt{\left(\frac{\Delta Q}{Q}\right)^2 + \left(\frac{\Delta A}{A}\right)^2 + \frac{1}{4}\left(\frac{\Delta H_1}{H_1}\right)^2} \tag{5}$$

where $\Delta A/A$ and $\Delta H_1/H_1$ refer to the relative error of $A$ and $H_1$. Both $\Delta A/A$ and $\Delta B/B$ are geometrical parameters and reply on the manufacture accuracy, while $\Delta Q/Q$, $\Delta H/H$ and $\Delta H_1/H_1$ depend on the measurements.

*6.3. Model Scale and Other Factors*

Depending on the test purpose, most of our models are constructed within the scale range $\lambda = 40$–$60$. It is an established point of view that a model within the range should provide equivalent results. About 65% of the present models are built $\lambda = 50$; a limited number have scales $\lambda = 30$ and

100. If the laboratory space and pumping capacity are sufficient, a larger model is preferred. The extra costs are often marginal.

A spillway, either of a surface or bottom type, is the control structure; its geometry is the major source of errors in modeling. Attention is paid to bottom outlets with small dimensions; a large model should be built to avoid significant scale effects.

Spillways in the past tests were constructed with either Plexiglas, hardwood, sheet steel or their combinations. Most recent model spillways are manufactured with the machine milling or 3D printing. By doing so, the construction error is reduced.

Scale effects of Reynolds and Weber numbers (R, W) are always an issue of discussion in spillway discharge modeling [23,24]. $R = (uh)/v$ and $W = u/(\sigma/\rho h)^{0.5}$, where $u$ = flow velocity, $h$ = characteristic length of flow, $\rho$ = water density, $\sigma$ = surface-tension coefficient and $v$ = water kinematic viscosity. Figure 5 gives the prototype and model of Bergeforsen's new spillway (Table 2). The model scale is $\lambda = 50$.

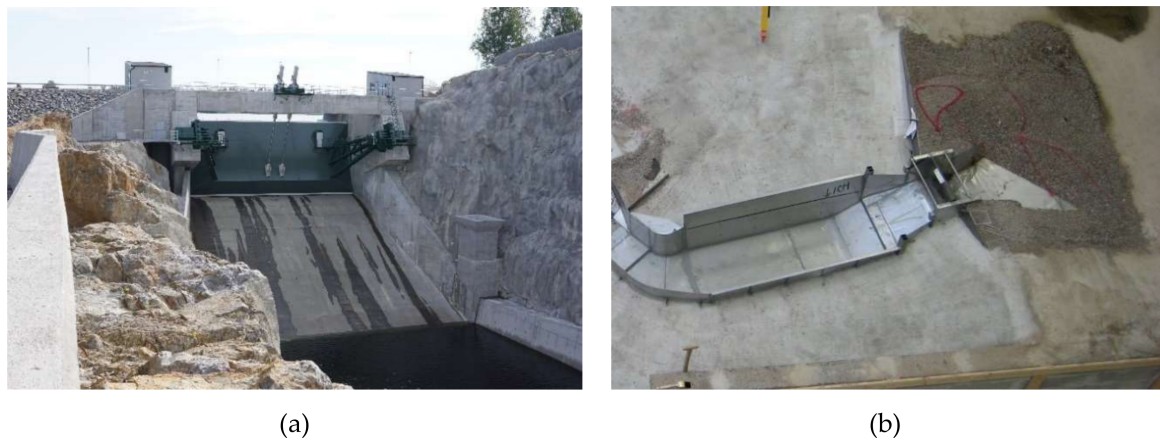

(a)          (b)

**Figure 5.** Bergeforsen spillway: (**a**) Prototype (© Vattenfall); (**b**) Scale model, $\lambda = 50$.

Figure 6 showcases, along the spillway opening centerline in the model, the streamwise R and W changes. For comparison, a hypothetical scale, $\lambda = 30$, is also plotted. The x-coordinate is positive in the flow direction, with $x = 0$ at the upstream dam front. The negative values correspond to the reservoir area. $B = 15$ m, $H = 10.25$ m, and $Q \approx 1500$ m$^3$/s at the DRWL (Table 2) [5,9,25]. Differences in both R and W exist between the scales. For the same $\lambda$, the R and W values are low in the reservoir area and become higher when water enters the spillway opening. In relation to the spillway flow dimension (*B* and *H*), the reservoir water area is always larger, thus leading to much lower *u*, R and W values, especially further upstream. This implies that, for discharge modeling, the viscous and surface forces always play a role, probably not significant, in the results. Attention should also be paid to shallow reservoirs and small discharges. A surfactant, detergent or equivalent solutions can be used to reduce the surface-tension effect.

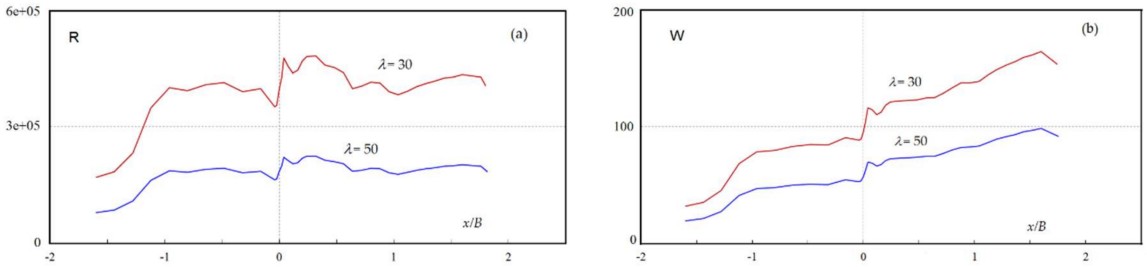

**Figure 6.** Bergeforsen spillway model; (**a**) R diagram for $\lambda = 30, 50$; (**b**) W diagram for $\lambda = 30, 50$.

### 6.4. Measurements of Flow Rate

A sharp-edged triangular or rectangular overflow weir was used in most of the past model tests for measurement of flow rates. One can speculate that such a weir often involved manual reading and might introduce errors. A poorly installed weir could lead to an error as large as ±5%. In the present tests, a magnetic flow meter, sometimes two, is always adopted. A calibrated magnetic meter has a higher accuracy, usually with a relative error lower than ±(1–2)%. It is not surprising that magnetic meters can also lead to large measurement error if not properly calibrated.

## 7. Numerical Modeling

CFD modeling is performed either separately or in combination with the present physical model tests. In Sweden, however, its use has not come to the same extent as the physical model tests. As a sole tool to solve such issues as energy dissipation, cavitation, and air entrainment, CFD involves aspects that are still topics of research. As for prediction of the spillway discharge capacity, experiences and confidence are gained in connection with the dam refurbishments.

### 7.1. Comparison between CFD and Model Tests

The spillway structure of the Torpshammar dam is a typical case, for which both model tests and CFD modeling are performed along with each other. Its layout is given in Figure 7, featuring two overflow gates and two bottom outlets. All the gates are of upward-going segment type. The width of each opening is 6.0 m (Tables 2 and 3). Two horizontal beams exist upstream of the spillway to maintain the long-term stability of the sidewalls holding the embankment dam. Their intrusions in the water disturb the approach flow, complicate the flow pattern, and negatively affect the discharge capacity. The physical model and the CFD modeling result are shown in Figure 8. The latter refers to the flow pattern with all four openings fully open for discharge at the DRWL. The solid lines correspond to the interfaces between water and air.

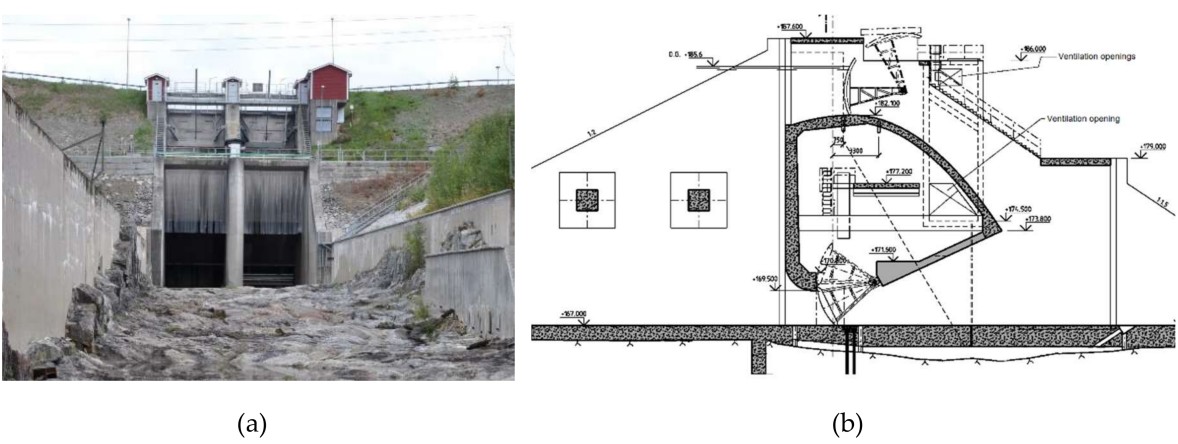

(a)                                           (b)

**Figure 7.** Torpshammar's gated overflow openings and bottom outlets: (**a**) Looking upstream from the discharge channel; (**b**) Longitudinal profile.

Between the CFD ($Q_{cfd}$) and model tests ($Q_{model}$), the relative difference in discharge is defined as

$$\beta(\%) = (Q_{cfd} - Q_{model})/Q_{model} \tag{6}$$

in which $Q_{model}$ (m$^3$/s) refers to either $Q_{present}$, $Q_{past}$ or both. In a rehabilitation project, CFD simulations are usually performed for different options. However, listed here are only those that have the same geometrical layout and approach flow conditions as the model tests. In most cases, the simulations are based on the echo-sounded bathymetries as used in the present model studies. Table 5 compares the results for 8 dams, with 7 surface spillways and 2 bottom outlets [5,7,25–31].

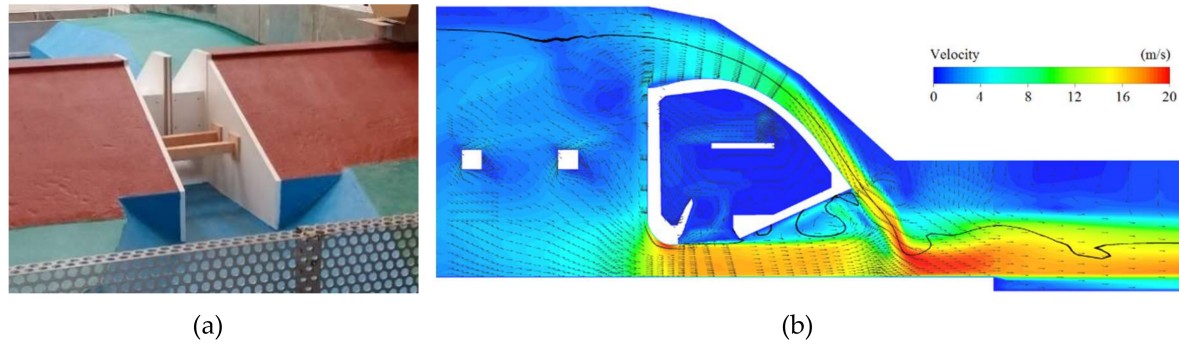

(a)　　　　　　　　　　　　　　　　　　　　　(b)

**Figure 8.** Torpshammar spillway: (**a**) Present physical model ($\lambda$ = 30), upstream view; (**b**) CFD results showing the flow pattern at simultaneous operation of all four openings.

The $\beta$ values are between $-3.8$ and $+2.9$. In comparison with the test results, the simulations give an either slightly higher or lower value. For several cases, they are very close to each other. It should be noted that the present model tests are conducted at Vattenfall R&D by its own personnel, while the CFD simulations are performed by persons from another institution, often at other occasions than the model tests. The CFD user do not in advance have access to the model test data.

**Table 5.** Prediction of spillway discharge, comparison between CFD and physical tests.

| Dam Name | Physical Model Tests | | CFD Simulations | | $\beta$ (%) |
|---|---|---|---|---|---|
| | Year | $Q_{model}$ (m³/s) | Year | $Q_{cfd}$ (m³/s) | |
| Bergeforsen (new spillway) | 2010 | 1512 | 2016 | 1505 | $-0.5$ |
| Höljes | 2008 | 1678 | 2013 | 1715 | 2.2 |
| Gallejaur | 1962 | 724 | 2017 | 720 | $-0.6$ |
| | 2004 | 720 | | | 0 |
| Långströmmen | 2015 | 475 | 2015 | 485 | 2.1 |
| Rusfors | 1962 | 1625 | 2016 | 1598 | $-1.7$ |
| Storfinnforsen | 1950 | 453 | 2015 | 436 | $-3.8$ |
| | 2008 | 449 | | | 2.9 |
| Torpshammar (2 surface gates) | | 135 | | 128 | $-3.7$ |
| Torpshammar (2 bottom outlets) | 2017–2018 | 485 | 2018 | 494 | 1.9 |
| Torpshammar (all gates) | | 605 | | 607 | 0.3 |
| Vatnsfell (Iceland) | 1999 | 350 | 2007 | 347 | $-1.0$ |

In Table 5, all the results refer to the spillway discharges at the full gate openings at the DRWL. CFD predictions are in some cases performed for discharges at varied gate openings. One typical case is for the bottom outlet at Storfinnforsen (39-m high buttress dam). Figure 9a shows its configuration ($a$ = 3.65 m, $b$ = 5.50 m, Table 3). It is equipped with an upward-going segment gate. Irrespective of tailwater levels, the outflow from the gate is free. Figure 9b shows the water-air flow pattern along the outlet centerline, at the opening 2 m at the DRWL. The comparison between the model tests and CFD is summarized in Table 6. The maximum relative error is 2.9%. Even at varied gate openings, CFD reproduces reliable discharge results.

**Table 6.** Storfinnforsen bottom outlet, comparisons of $Q_{model}$ and $Q_{cfd}$ at fixed gate openings.

| Gate Opening (m) | $Q_{model}$ (m³/s) | $Q_{cfd}$ (m³/s) | $\beta$ (%) |
|---|---|---|---|
| 1.00 | 106 | 108 | 1.9 |
| 2.00 | 200 | 203 | 1.5 |
| 3.00 | 306 | 310 | 1.3 |
| 3.65 | 449 | 436 | $-2.9$ |

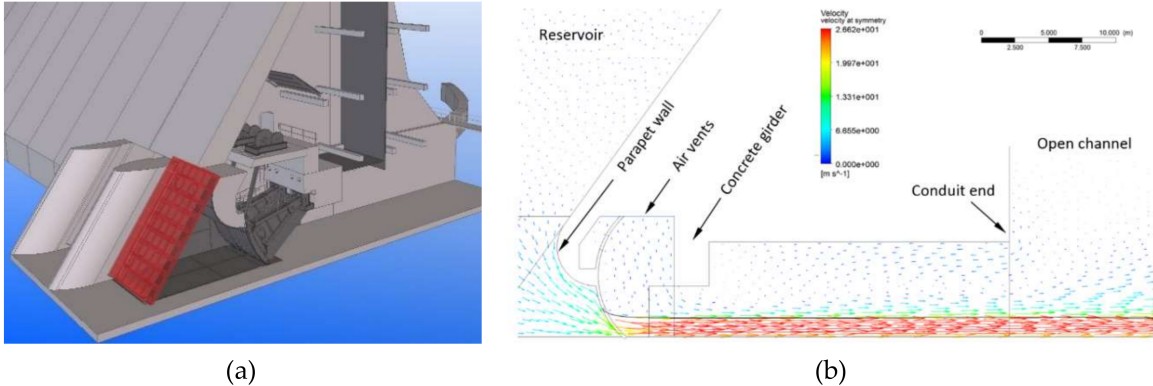

(a)　　　　　　　　　　　　　　　　　　　　　　　　(b)

**Figure 9.** Bottom outlet at Storfinnforsen dam: (**a**) Layout; (**b**) CFD results of two-phase flow.

*7.2. Sources of Errors in CFD*

For CFD modeling, simulations are usually made for the prototype dimensions. The choice of the upstream area should follow the same principle as for the scale models. The sources of errors include the following issues [32–36]. The discussion here does not intend to be all-inclusive; it touches only on a few major aspects.

General CFD codes are used in the projects for the spillway discharge modeling. These include FLUENT and CFX in the ANSYS package (Ansys Inc., Canonsburg, PA, USA). The modeling involves both air and water; this is the reason a two-phase model, such as the Volume of Fluid (VOF) and the Two-Fluid Model, is always used. The RNG *k-ε* turbulence model is found to be superior to other turbulence models and is used for the free-surface flow simulations [37,38]. A recent review of modeling techniques related to water-air flows in hydraulic jumps is made by Valero et al. [39] and Viti et al. [40].

It is essential to have a high-quality grid to achieve reliable numerical results. Discretization error and grid independence must be checked for the grid to be used. A higher mesh density is given to near-wall regions and areas of interest including the free surface and the spillway opening.

The numerical scheme affects modeling accuracy. Even for steady-state problems, a time-dependent procedure is usually applied for the solution. For an explicit scheme, the Courant criterion should be followed. For an implicit formulation, there is no stability requirement for the time step. It is however necessary to set the time step at least one order of magnitude smaller than the smallest time constant in the system. At each time step, to observe the number of iterations needed to converge is a proper way to judge its choice. The ideal number is 5–10 iterations. With more iterations, the time step is too large and vice versa.

For the reservoir boundary conditions, the hydrostatic pressure distribution is a better choice for the approach flow; the inflow distribution along the boundaries is thus adjusted in the light of pressure and velocity coupling. Specifying a constant flow velocity is a less realistic choice and would require a longer distance for the flow to adjust itself. If the reservoir area is not sufficiently large, errors in discharge would simply occur. For the downstream boundary, the domain should end up with a cross-section of supercritical flow on the chute, with an outflow condition.

The user should have a solid physical background of spillway design and follow the best-practice guidelines for free-surface modeling in terms of reservoir size, grid density independence, choice of turbulence model, boundary conditions, convergence criteria etc. For more details, one should refer to e.g., Celik et al. [32], ANSYS Inc. [37] and ERCOFTAC [41,42]. A description of the development and evaluations of CFD studies is given in Blocken and Gualtieri [43].

If CFD modeling aims only to predict the flow discharge, the numerical model suffices to include a short length downstream of the gate, as long as the downstream boundary does not affect the outflow. In some cases, the discharge results are extracted from more complex flow simulations involving energy dissipation and air entrainment. A general description is found in [44]. A few recent studies on

the topic include Teng and Yang [28], Damion [45], Valero and García-Bartual [46], Bayon et al. [47], Wang et al. [48], Valero et al. [49] and Yang et al. [50].

## 8. Concluding Remarks

Most hydropower dams in Sweden were built during 1945–80. Compared to the present design-flood criteria, most spillways were undersized. The current dam-safety guidelines stipulate that the spillway structures should be upgraded to accommodate the higher design floods.

The determination of spillway discharge capacity is a major issue for both the original design of a dam and its later refurbishment. In conjunction with the upgrading, many existing dams are re-tested in the laboratory. In some cases, CFD modeling is also performed. This gives us the possibility to compare the discharge data of the existing spillways between the model tests performed a few decades apart and even with the CFD simulations if available.

More than 20 dam spillways are included, of both surface and bottom types. For some dams, the deviation is as large as 11.2%. The reason for the discrepancy is presumably due to the differences in model construction quality and flow measurement method. The recent use of the milling technique and 3D printing reduces the source of errors in model construction and improves the model accuracy in terms of surface finishing.

Notwithstanding the rapid development of CFD, the most effective solution of dam hydraulic issues involving heuristic engineering reasoning in the face of constraints, design, and prototype implementation, is still best handled by the intelligent use of physical hydraulic models. In some countries, but not in Sweden, physical models are usually required by the governmental authorities, which is one of the reasons why the transition towards CFD is slow. For the spillway discharge modeling, the issue is less complex and CFD should provide reliable results equivalent to the model tests, as long as the best practice guidelines are followed by the user. Of course, the optimum is a combination of both techniques in seeking the solution.

**Author Contributions:** J.Y. outlines the study and performs the investigation, with participation from Q.X., P.T. and P.A. Part of the CFD results is based on the simulations performed by P.T. The manuscript is written by J.Y., with comments from the other authors.

**Funding:** The study is carried out as part of the dam-safety program within Vattenfall's R&D Hydro portfolio 2018.

**Acknowledgments:** Part of the physical model results is presented at HydroVision International, June 2013, Denver, USA (©HCI Publications). J.Y. is grateful to Dr. Christian Bernstone and Dr. Mats Billstein of Vattenfall R&D for project coordination. Any opinions and views expressed here are only those of the authors and are not the views of or endorsed by the institutions or dam owners involved.

**Conflicts of Interest:** The authors declare no conflict of interest.

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
