# Peer review of "The Past and Present of Discharge Capacity Modeling for Spillways—A Swedish Perspective"

_fluids, doi:10.3390/fluids4010010_

Round 1
Reviewer 1 Report
General comments: Very interesting paper with useful information and investigations relevant for design of new spillways and reconstruction of existing spillways. Well worth publishing. The reviewers only complaints are some minor language details, and a few question to the content. Comments and questions are given below.
Line 15: "Indicated” is a too vague statement. “Resulted” is perhaps better.
Line 17: “The paper briefs” can be changed to “This paper presents”.
Line 26: The information that the CFD simulations are conducted without knowlegde of the physical model results in advance can be given already in the introduction. That is a crucial piece of information.
Line 29: Remove “accuracy”. Not a necessary keyword.
Line 37: “Effect” must be changed to “power” or “installed capacity” to follow hydropower standards.
Figure 1: The caption in the figure states “annual power addition (TWh)”. If it is truly power use TW, and if it is energy (TWh) change the text to "annual energy addition".
Line 63: The spillway discharge capacity was perhaps not underestimated (would mean that the physical model test were unaccurate), but the necessary spillway discharge was underestimated (meaning that the hydrological method to determine the design flood was unaccurate). This is a problem throughout the paper, it is written in a way that one cannot clearly understand what causes many of the spillways to have insufficient capacity today; is it because of new dam safety guidelines, or is it because of unaccurate physical modelling in the past, or is it because the hydrological methods to determine the design flood was unaccurate. It is in most of the cases a combination of the two last causes as the paper demonstrates.
Line 100: Use “doubling of the width” instead of “100% widening”.
Line 102: “Statistics has been conducted on”.
Line 116: “In light of” is a vague statement. Perhaps “based on” is better.
Line 116: Missing the word “new” before “flood determination criteria”?
Line 147: Scrutinized to examine is a wierd statement as scrutinized means examine closely.
Section 6.2: Missing a discussion on whether “C” in the overflow equation can be treated like a constant in this case. “C” is dependent on many variables, such as depth and crest geometry.
Line 241: Euler number (relation of pressure to inertia forces) is relevant for bottom outlet gates. For water, where density is almost similar in the model and prototype, the Euler number and Froude number is scaled equally. I have not seen any research on the influence of water density and the Euler number for physical scale modelling, which probably means that no one think it has a large effect ?
Line 246: Not state in table four that there has been a model test with scaling factor 30 of Bergeforsen?
Line 266: Usually physical models are required by the governmental autorities, and that is one of the reasons the transition towards CFD is slow.
Line 283: Remove “the” in “all four the openings”.
Line 314: VOF means “volume of fluid” not “volume of fraction”.
Line 315: List referece to where the RNG k-e turbulence model is found to be superior.
Line 317: “It is essential to have a high quality grid”.
Line 339: Add “compared to present dam safety guidelines”. Probably many of the spilways were not under-sized according the dam safety guidelines that were available at the time of construction. Now it seems they were bad engineerings in the past.. my guess is that they were not ?
Author Response
The reply is attached.

Reviewer 2 Report
Please, carefully consider my comments in the attached manuscript.

Author Response
The reply is attached.

Round 2
Reviewer 2 Report
I am satisfied with the improvements done by the Authors. I think that the manuscript can be published on Fluids.